# Research

palaeontology, evolution, bioinformatics

cheilostome bryozoans, fossil occurrences, palaeobiodiversity, natural language processing, information extraction, literature compilation

**Authors for correspondence:**
Bjørn Tore Kopperud
e-mail: b.t.kopperud@nhm.uio.no
Lee Hsiang Liow
e-mail: l.h.liow@ibv.uio.no

# Text-mined fossil biodiversity dynamics using machine learning

Bjørn Tore Kopperud[1], Scott Lidgard[2] and Lee Hsiang Liow[1,3]

[1]Natural History Museum, University of Oslo, PO Box 1172, Blindern, 0318 Oslo, Norway
[2]Integrative Research Center, Field Museum, 1400 South Lake Shore Drive, Chicago IL, 60605, USA
[3]Centre for Ecological and Evolutionary Synthesis, Department of Biosciences, University of Oslo, PO Box 1066, Blindern, 0316 Oslo, Norway

BTK, 0000-0002-7360-7087; SL, 0000-0002-0446-4705; LHL, 0000-0002-3732-6069

Documented occurrences of fossil taxa are the empirical foundation for understanding large-scale biodiversity changes and evolutionary dynamics in deep time. The fossil record contains vast amounts of understudied taxa. Yet the compilation of huge volumes of data remains a labour-intensive impediment to a more complete understanding of Earth's biodiversity history. Even so, many occurrence records of species and genera in these taxa can be uncovered in the palaeontological literature. Here, we extract observations of fossils and their inferred ages from unstructured text in books and scientific articles using machine-learning approaches. We use Bryozoa, a group of marine invertebrates with a rich fossil record, as a case study. Building on recent advances in computational linguistics, we develop a pipeline to recognize taxonomic names and geologic time intervals in published literature and use supervised learning to machine-read whether the species in question occurred in a given age interval. Intermediate machine error rates appear comparable to human error rates in a simple trial, and resulting genus richness curves capture the main features of published fossil diversity studies of bryozoans. We believe our automated pipeline, that greatly reduced the time required to compile our dataset, can help others compile similar data for other taxa.

## 1. Introduction

How have scientists determined the history of biodiversity on our planet? The radiations of unicellular organisms, plants and animals, rates of diversification and extinction, correlation of past biodiversity levels with environmental forcing factors, mass extinctions and recoveries—all of these and more are ultimately reliant on, or at least calibrated by, published occurrences of fossil taxa and their geologic ages. For more than 200 years, palaeontologists have used their own work and mined the work of others to document taxonomic richness in successive geologic time intervals [1]. This effort has historically involved searching out and manually compiling published occurrences of lower-level taxa (e.g. species, genera, families) within a given higher-level group (e.g. order, class, phylum) and then inferring the changing patterns of richness [2,3].

Despite considerable progress in statistical methods that aim to compensate for occurrence gaps and known biases of the fossil record [4–8], we are still some ways away from a comprehensive understanding of the history of global biodiversity. One of the foremost impediments remaining is the costly, labour-intensive business of extracting fossil occurrence data from a continually expanding, scattered literature, even for palaeontological experts analysing one or another moderately taxon-rich group. Depending upon the extent and availability of previous fossil diversity studies on that group, this manual process could take months or even years to complete. Moreover, the methods and sources that underpin such compilations may or may not have been recorded and made available [9,10], potentially compromising scientific repeatability.

Hence, community efforts have built large public data compilations of taxonomic nomenclature or taxon occurrences, for instance, the Global Biodiversity Information Facility [11], World Register of Marine Species [12] and the Paleobiology Database (https://paleobiodb.org/). While one million records entered over 20 years and 300 credited publications attest to the impact of the Paleobiology Database, key data gaps persist and require sustained data gathering and input.

A promising approach to bridging data gaps is to partially automate the process of populating such data compilations, using information extraction techniques. Information extraction using natural language processing has become a mainstay in the biomedical field [13]. For instance, drug–drug interactions and mental illness symptoms are presented in new publications every month and such data extracted [14,15] from disparate journals can be used to further our understanding of human diseases and treatments. Text-mining approaches have also begun to impact knowledge discovery in biodiversity studies of living organisms [16,17]. With respect to palaeontological biodiversity studies, Peters *et al.* [18] pioneered automated information retrieval of fossil observations and their geologic time intervals. Here, we improve the pipeline proposed by Peters *et al.* [18]. We apply natural language processing tools [19,20] that aid in information retrieval by supervised learning [21–23]. Advantages of such a text-mining machine-learning approach include (i) enhanced efficiency in terms of time expended to find and retrieve data from the primary literature and the costs thereof, (ii) the ease of error estimation, and (iii) repeatability and transparency in terms of the source and nature of the data.

We show how our pipeline can be applied to organismal groups, especially those in which data are wanting in public databases, yet abundantly available in scattered publications. One such group is Bryozoa, a phylum of colonial animals in freshwater and more commonly in benthic marine environments. They are habitat constructors that enhance the biodiversity of other organisms [24], are instrumental in oceanic organic filtering and are major biogeochemical carbonate engines in the global carbon cycle [25]. Their rich fossil record and complex morphologies have also been critically important in testing macroevolutionary theory [26]. Despite several reviews of their fossil biodiversity history [27–29], publicly available data are lacking in a form that is amenable to relevant statistical analyses [5,8,30–32]. Our aim here is to build an occurrence database of observations of fossil cheilostomes (order Cheilostomatida [33]). The cheilostomes are the most species-rich group of bryozoans for which there are currently about 4800 known extant members [29]. We explicitly quantify both human and machine error in retrieving taxon names and their time intervals of occurrence. We then compare the inferred history of cheilostome bryozoans using our machine-read data to their established palaeobiodiversity pattern [29] and find that major features of their diversity changes are recovered across their 150-million-year history. Finally, we discuss the pros and cons of semi-automated text-mining techniques and suggest avenues for future improvements.

## 2. Material and methods

### (a) Data sources

Public compendia such as Web of Science, Zoological Record and Google Scholar are primary sources for scientific publications from major publishers, especially those with a modern Web-presence. However, many key taxonomic publications appear in more obscure platforms such as museum publications, conference volumes, theses and governments reports, where accessibility through online sources is often limited. For this reason, we obtained bibliographic collections from experienced scientists working on bryozoans for much of their careers, including one of us (SL). PDFs (Portable Document Format) of publications that were likely to contain relevant species occurrence data from the Jurassic through the Holocene (including those in [27,28]) were then combined. This corpus consisted of over 10 000 bibliographic references, only some of which were likely to contain fossil species occurrence data. We filtered those with accompanying PDFs, yielding more than 2000 documents in the English language; more than 800 of these contained fossil occurrences that were useful for our text-mining pipeline (see below).

Recognizing words as bryozoan taxa or geologic time intervals entails reference lists of names. For bryozoan taxon names, we used the World Register of Marine Species [12] and the online taxonomic compendium of an experienced bryozoologist [34]. For geologic time-interval names, we used Macrostrat [35] and GeoWhen [36]. Publications documenting fossil occurrences include taxonomic monographs, systematic treatments and faunal lists. These publications contain formal taxonomic descriptions of fossils, discussions of the described or previously described taxa, taxon lists, summary tables or illustrations of taxa. For our text-mining procedure, we focused on mentions of taxonomic names in full sentences. In other words, we ignored lists, tables and figures, as they are more specialized data formats for which methods in information extraction are not yet as well-developed. We discuss the consequences of not compiling these latter data for taxonomic richness in later sections.

### (b) Overview of information retrieval

The end-product that we seek is a dataset of observations of cheilostome fossil taxa and their geologic ages and corresponding bibliographic references. To do so, we used Poppler (https://poppler.freedesktop.org/) to extract plain text from the PDFs and then subject this text to information retrieval, as summarized in figure 1. There are three main steps in our information retrieval procedure. First, we used automated approaches to isolate sentences in which taxa and age names co-occur and produced data consisting of 'candidate' taxon-age pairs and the sentences in which they occur. Second, we manually labelled a subset of these candidates for constructing a set of 'gold labels' which are used in training a machine-learning classifier. Third, we applied the machine-learning classifier to the entire set of the candidates and generated our fossil occurrence data. We briefly detail each of these steps in the next sections.

### (c) Sentence parsing and named entities

We used Stanford CoreNLP [37] to automate the annotation of linguistic information, such that we can recognize 'tokens' (i.e. words and punctuation), starts and ends of sentences, and named entities, specifically bryozoan fossil taxa and geologic time intervals. Taxon names are Latinized and can be identified in part by word morphology, but also via in-sentence context and grammar. In other words, we could have set up named-entity recognition using a machine-learning classifier [38]. However, it is next to impossible to distinguish among names from different taxonomic groups based on grammatical context and word morphology alone. Hence, we chose to use a rule-based approach, not least because a nearly exhaustive list of post-Palaeozoic bryozoan Linnaean binomials (including all cheilostome bryozoans, our target group) is already available. We used this list and our compiled list of geologic age interval

**Figure 1.** (a) The general workflow for automatic information extraction of fossil occurrence data. (b) The machine-learning classifier. A bidirectional long short-term memory (LSTM) recurrent neural network, with the first example candidate as input. The numbers given are illustrative. Dashed arrows indicate dependency grammar links. See electronic supplementary material for details on the classifier. The figure style is inspired by Miwa & Bansal [23]. (Online version in colour.)

names to generate a set of TokensRegex expressions [39] that were matched with occurrences within sentences (i.e. named-entity recognition, see electronic supplementary information for details). For one or more consecutive tokens that indicate one entity (e.g. 'Bugula', 'Setosellina roulei', 'lower Miocene'), we use the term 'span'. A candidate refers to a pair of spans, one of which is a full Linnaean binomial or genus name, and the other a geologic age. We used the Snorkel framework [40] to locate candidates, treating each candidate as independent information. The automated workflow thus far resulted in a product consisting of candidate pairs and the sentences in which they co-occur.

## (d) Gold labels: manually labelled candidates

The set of candidates from the previous section consists of co-occurrences of a name and age, but these can be either positive or negative candidates. An example of a positive candidate (from [41], p. 54), where it is explicitly stated or strongly implied that the species occurred in the given geologic time interval, is:

Remarks—A few, small, infertile colonies of **Setosellina cf. roulei** have been found encrusting the undersides of very thin platy corals from the [late Burdigalian] and the **Serravallian**.

Bold font indicates the relevant spans, and square brackets indicate spans that are not currently under consideration. An example of a negative candidate (from [42], p. 419) is:

... fix the identity of **Cribrilina punctata**, the type species of the genus [Cribrilina], itself the type genus of the cosmopolitan **Cretaceous** to [Recent] family Cribrilinidae.

The candidates could be manually labelled as positive or negative for downstream use as part of a fossil occurrence dataset. By automating this task with a machine-learning classifier, we can more easily annotate large volumes of data, tackle new sources of data (e.g. newly published articles or old ones made available) or apply the classifier to other groups of taxa. The machine-learning classifier learns from examples (a supervised learning approach, see the next section). Two human annotators manually labelled 10 416 candidates, 1000 of which were labelled by both persons. These annotators were not taxonomists or bryozoologists, but had basic degrees in biology, understood the Linnaean classification system and had geological knowledge sufficient to recognize stratigraphic age units. For candidates where the annotators disagreed, we assigned a positive or negative label at random (see electronic supplementary material). These resulting human-annotated data are our labelled candidates, each with a corresponding gold label.

This labelling of candidates in the context of retrieved sentences takes much less time than an approach that starts with reading through numerous papers or monographs to find the relevant data wherever it might occur in the text and subsequently entering each separate taxon name and age.

## (e) Machine-learning classifier

In order for a machine-learning classifier to retrieve the semantic relationship between two spans in a sentence, we characterized features of the sentence's elements. We described the links among words [19] using pre-trained machine-learning models for English dependency grammar in CoreNLP [43]. We used the dependency grammar tree to compute the shortest dependency path (SDP) between the two spans [22,44], as illustrated in electronic supplementary material, figure S4. The resulting SDPs from the labelled candidates and their gold labels were then used to train a supervised machine-learning classifier (figure 1b), specifically a Long Short-Term Memory recurrent neural network or LSTM [45] implemented similarly to Xu et al. [22].

To summarize, our neural network consists of three layers: a word embedding layer, a bidirectional LSTM layer and a third hidden layer (figure 1b). The output of the third and final layer is a vector of length two, which represents the probability mass for our relation classification task. We used Keras [46] to implement this neural network (see electronic supplementary information for details).

We split the labelled candidates into training, validation and test sets (80, 10, 10% of the labelled candidates, respectively). The training data were used to fit the parameters of the classifier. The validation data were used for two purposes, to decide when to stop training, and to hand-tune the hyperparameters (e.g. the layer dimensions and learning rates, see electronic supplementary information for details). We used the test set to evaluate classifier performance. We employed metrics that are standard in binary classification problems [47] to evaluate model performance: accuracy (the ratio of correct predictions to all predictions), precision (the ratio of true positive predictions to all positive predictions), recall (the ratio of true positive predictions to all positive labels), false positive rate (FPR, the ratio of false positive predictions to all negative labels) and F1 (the harmonic mean of precision and recall). In classification problems such as ours, a decision boundary divides the decision space into categories. Here, the outputs of our classifier are in the interval [0,1], and we set our binary decision boundary (b) to 0.5 unless otherwise stated. If we assume that the test set is representative of the unlabelled candidates, the performance of the classifier on the test set will

Proc. R. Soc. B 286: 20190022

approximate how well it classifies the unlabelled data. As an indicator of the estimation error of the above-mentioned metrics, we fitted the neural network 100 times with the same hyperparameters and report their mean $\pm$ s.d. evaluated on the test set. To avoid confounding effects of similar sentences in a given document, we ensured that all candidates found in a single document were contained within only one of the sets used for training, validation, and testing. We used the relation classifier to automatically label remaining candidates. As there were some duplicates among these remaining candidates, we estimate that there were about 10 000 unique unlabelled ones.

## (f) Cheilostome palaeodiversity
In order to compare text-mined taxonomic richness through geologic time with previous publications, we considered only genera, even though we also text-mined full binomials. After the relation extractions of genera or binomials and geologic ages, we performed several steps of post-processing. We used Macrostrat [35], supplemented in a few cases by online lookups, to obtain the minimum and maximum ages of each named time interval. As genus names are commonly abbreviated (e.g. 'C. punctata') after their first mention in a text section, we performed a de-abbreviation step by doing a lookup of genus names in the previous 15 sentences, that begin with the capitalized letter in the abbreviation. The abbreviated genus names were successfully de-abbreviated in about 80% of the candidates without a full genus name (electronic supplementary material, figure S2). Biodiversity compilations must account for invalid names, synonymizations and related historical artefacts of the literature, reflecting taxonomic revisions. Our cheilostome genera were standardized and revised to valid ones as far as possible using WoRMS [12] and data from two of its primary contributors and editors—Bock [34] and Gordon (Dennis P. Gordon, personal communication, 2018), who is maintaining the most recent working document of the Treatise on Invertebrate Paleontology, Part G: Bryozoa [48]. Species-level synonymies are not treated here.

# 3. Results and discussion
## (a) Gold label annotation repeatability
We estimated an accuracy of 84.1% between the decisions made by the two human annotators for the mutually examined gold labels. This inter-annotator accuracy is a joint measure of noisy data, language ambiguity, human error and degree of conservatism by the annotators. Examples of noisy data include tables parsed as paragraphs, incorrect sentence boundaries, errors in optical character recognition and incomplete sentences such as those found in figure captions. We interpret the annotator repeatability as a workable baseline when judging the performance of the machine-classifier. Similar inter-person accuracy estimates have been reported in analogous human error assessments made by Peters et al. [18]. We also note that human error rates in data compilations are rarely explored quantitatively and never modelled in palaeobiological analyses on large datasets compiled from the literature (e.g. [49,50], but see [51] for an exceptional investigation of such errors).

## (b) Classifier performance
The relation classifier applied to our test set achieved a recall of $88.0 \pm 1.5\%$, accuracy of $82.2 \pm 0.8\%$, precision of $82.8 \pm 1.3\%$ and an F1 of $85.3 \pm 0.6\%$. Figure 2 illustrates the

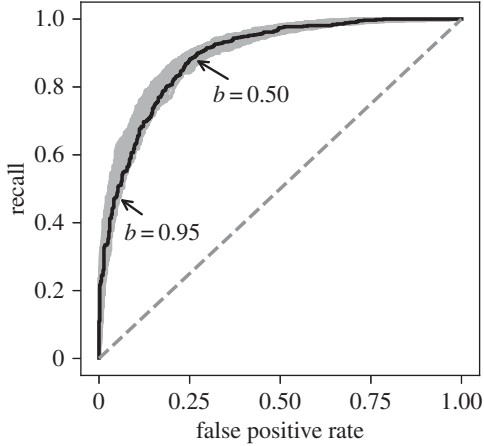

**Figure 2.** Receiver operating characteristic curve. The rates for the relation classifier are evaluated on the test set. Ninety-nine iterations are plotted in grey, and the one in black is chosen at random. $b = 0.50$ is the standard decision boundary, and $b = 0.95$ represents false positive rate of 5%. The area under the black curve is 0.90. The dashed line represents the expected rates given a random classifier.

trade-off between recall and the occurrence of false positives. The classifier accuracy at 82.2% is comparable to our inter-annotator labelling accuracy at 84.1%. Since human and machine accuracies are comparable, yet both far from perfect, we suggest that machine-reading performance is more limited by the ambiguousness of the data and annotator scoring ability than by the classification algorithms. In other words, repeatable candidate labelling is probably a major bottleneck for accurate machine-based relation classification. Whether the types of machine-classification errors overlap substantially with human-annotator errors remains an open question. While we do not know if our inter-annotator accuracy is representative of the human accuracy involved in manually populating similar knowledge databases (e.g. the Paleobiology Database), we have no reason to believe that repeatability in manual data-entry is substantially better or worse.

Incomplete fossil preservation, biased sampling and selective reporting of data in publications can all contribute to gaps in compiled databases. However, there are diverse approaches that model sampling probabilities or account for sampling incompleteness in the estimation of species richness [52,53] or diversification dynamics [4,5,7,31]. False negatives from machine-read texts are analogous to the above-mentioned gaps in compiled databases in terms of our ability to alleviate their effect in inferring past richness patterns and dynamics. False positives, however, falsely inflate estimates of taxonomic richness and we do not currently have modelling approaches to deal with this bias. To minimize the impact of false positives in our text-mined data, we constrained the predictions to have a relatively low FPR while maintaining a relatively high recall, as shown in the next section.

## (c) Genus richness counts
We used our text-mined predictions as well as the list of extant genera from the World Register of Marine Species [12] to produce a joint dataset from which range-through cheilostome genus richness is determined for successive time intervals over the past 160 Myr (figure 3). We set the decision boundary (b) to obtain a balance between relatively

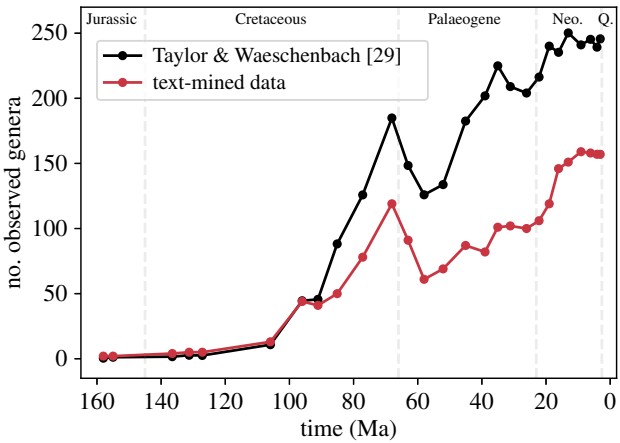

**Figure 3.** Range-through genus richness for cheilostomes. The curve from Taylor & Waeschenbach ([29], fig. 12) was obtained using a plot digitizer [54]. Our richness counts are supplemented with extant observations from WoRMS [12]. We used bins that are comparable with the bins used by Taylor & Waeschenbach [29]. The false positive rate evaluated on the test set is 27%. The geologic ranges for all genera are detailed in the electronic supplementary material, table S2. (Online version in colour.)

low FPR (25.9%) and high recall (88.1%). The FPRs are estimated for the test dataset. The net FPR is smaller than what is reported in figure 3. This is because some portion of candidates overlap with the training data, for which the classifier is comparably overfitted (FPR = 8.2 ± 4.2%, recall = 97.8 ± 1.0% for training data). However, the range-through genus counts are similar for subsets of data with FPR between 0.25 and 1 (electronic supplementary material, figure S6), indicating that the false positives are mostly adding to 'internal occurrences' rather than contributing to geologic range extensions (see electronic supplementary information, error inspections). In a previous, more comprehensive error-correction experiment using trilobites [51], the authors found that even after extensive corrections to geologic ranges and taxonomic nomenclature, the impact on genus richness counts through time was negligible.

The relative changes in our text-mined cheilostome genus richness curve (figure 3) are similar to those in a recently published review of bryozoan history [29], fig. 12), although our range-through genus counts are currently underestimates of that work. Genus richness increased in the Cenomanian (100.5–93.9 Ma) and steepened through the late Cretaceous (100–66 Ma), with a second diversification beginning in the Eocene (56–33.9 Ma). These trends are evident in both curves, as is the sharp decline in genus richness at the K-Pg extinction event and the end of the Danian around 62 Ma [55]. An advantage of our newly acquired occurrence-based dataset is that we now have multiple observations for most genera, including explicit literature sources. Such an occurrence-based dataset is amenable to taxonomic updates by systematists and consequently, revisions to inferred histories regardless of the approach applied. More beneficially, it facilitates the use of modern approaches to estimate richness patterns and diversification rates while accounting for incomplete sampling and sampling heterogeneity. These extensive analyses are outside the scope of the current work. However, the pipeline presented here will aid in paving the way for a more robust and nuanced inference of bryozoan palaeodiversity.

## (d) Common problems

While there are substantial advantages to text-mining genus/species age-observation data, we recognize several avenues for improvement in this study. Other problems and avenues that we do not discuss may become relevant as developments in machine-reading approaches unfold, as this field is in its infancy concerning applications to biodiversity studies. First, there are substantial amounts of historical and recent literature in languages other than English, notably French, German, Spanish, Italian, Japanese and Russian. While the text-mining task is conceptually similar regardless of the language, natural language processing tools are more advanced and accurate for high-resource languages such as English and Chinese [13]. Specialized tools (e.g. named-entity recognition, dependency grammar and relation classifiers) are typically tailor-made for a specific language, meaning that adding other languages effectively multiplies the effort required for automatic information extraction.

Second, our text-mining pipeline is entirely reliant on grammatically structured, isolated, single (complete or incomplete) sentences that describe the taxa and their ages. We do not take into account contextual information, e.g. linguistic coreference [56] among sentences. Similarly, taxonomic treatments are an example of extreme context dependence, where genera or species typically are given in headlines followed by concise descriptions in the following paragraphs. Thus, many taxonomic descriptions are not based on grammatical information but rather the spatial layout of paragraphs and typefaces of words in the article—both of which are ignored by our approach. More specialized tools are required to process taxonomic treatments more effectively.

Third, we neglect information in tables and figures apart from their captions. Govindaraju *et al*. [57] demonstrated how combining natural language and tables could improve information extraction. However, this is conditional on a machine-readable data structure for tables, and table extraction from PDFs remains difficult to achieve with accuracy at this time [58].

A fourth problem is one of optical character recognition errors (e.g. incorrectly parsing a letter, or parsing words as letters separated by whitespace). Text normalization has been employed to handle similar problems in social media corpora [59,60]; however, normalization may also remove or distort information, and it remains to be tested on taxonomic literature. Correct parsing of words and paragraphs is inherently limited by design choices in the PDF standard. Publishers are becoming increasingly aware of this problem; for instance, Elsevier provides researchers with machine-readable plain text or XML (Extensible Markup Language) formats of their articles through their text and data mining initiative.

Any of these above-mentioned problems may contribute to our underestimations in comparison with Taylor & Waeschenbach [29], even though any data compilation of fossil occurrences will always be incomplete due to both the nature of the fossil record and the process of fossil recovery.

## (e) Outlier inspection

In addition to false negatives that may contribute to underestimations, we also have false positives, some of which are temporal outliers. While most genera have reasonable estimated time spans (73% under 25 Myr), there are some

genera that exhibit suspiciously long time spans (electronic supplementary material, figure S3). False positives that contribute to internal occurrences of taxon stratigraphic ranges are simultaneously less problematic but more difficult to remove from machine-read data. We inspect and discuss some of the stratigraphic outliers in the electronic supplementary material, figure S3. Some of the errors we found could have been avoided by setting a slightly more conservative FPR when sub-setting the data before analysis. Another type of false positive arises from taxonomic ambiguities. For instance, the genus *Callopora* is an accepted name for a Cenozoic cheilostome named by Gray [61]. However, we have mined data from Ernst & Nakrem [62] who referred to a historical mention of *Callopora* Hall [63] in their discussion of a Palaeozoic trepostome. Such an issue could in principle have been avoided by using author names for genera or species. However, author names are not always consistently supplied in-sentence and are often given in a different font or typeface, which is problematic for text parsing. In other words, knowledge of taxonomic practice in general and the taxonomy of the group in question is still crucial for an informed usage of machine-compiled data. We have in fact not dealt with species-level synonyms in this contribution because of the immensity of the task. However, past studies of fossil biodiversity have shown that taxonomic revisions do not necessarily change the nature of global scale patterns when datasets are sufficiently large [51,64].

## (f) Concluding remarks

During the course of this work, it was clear that publications containing useful information on relatively understudied organisms such as Bryozoa remain difficult to locate without expert knowledge. In order to understand both past and present biodiversity, we need continued training and support of taxonomic experts [65,66]. Automating some information retrieval tasks will save time and allow taxonomists to focus on other tasks where their expertise is essential, such as describing new species and revising old names.

We believe that the pipeline presented here can be easily adapted for other groups of organisms, likely without taxon-specific gold labels, to build and augment large knowledge-bases of fossil occurrences. While human-vetted databases may be superior taxonomically, they are expensive to curate and maintain, and their data sources are sometimes difficult to trace, unlike our text-mined data. Faced with a growing literature and diminishing person-power and funding, machine-reading approaches can complement vetted databases by targeting knowledge gaps and hence contribute to understanding large-scale biodiversity changes and evolutionary dynamics. The machine-read fossil occurrence data presented here captured major palaeodiversity patterns in cheilostome bryozoans, despite varied sources of error that are generally inherent in large databases.

Data accessibility. The labelled candidates, word embeddings, cheilostome genus ranges, genera synonyms, code for named-entity recognition, candidate extraction and the classifier are available on the Open Science Framework at https://osf.io/dn4gy.

Authors' contributions. B.T.K. and L.H.L. designed the study, B.T.K. did the analyses and all authors wrote the manuscript.

Competing Interests. We declare we have no competing interests.

Funding. This project was supported by the European Research Council (ERC) under the European Union's Horizon 2020 research and innovation programme (grant agreement no. 724324).

Acknowledgements. We thank Olja Toljagić for help with labelling the candidates, Björn Berning, Dennis P. Gordon, Paul D. Taylor and Phil Bock for assistance with taxon lists, Caroline Schuette, Björn Berning and Kamil Zágoršek for assistance with the bibliography, and Farhad Nooralahzadeh, Shanan E. Peters, Ian Ross and Trond Reitan for discussions. We also thank the two anonymous reviewers whose criticism helped improve the narrative of our manuscript.

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
