## [Reviewer comments · Proceedings of the Royal Society B: Biological Sciences]

Review History

RSPB-2019-0022.R0 (Original submission)

Review form: Reviewer 1

Recommendation

Accept with minor revision (please list in comments)

Scientific importance: Is the manuscript an original and important contribution to its field?

Good

General interest: Is the paper of sufficient general interest?

Good

Quality of the paper: Is the overall quality of the paper suitable?

Good

Is the length of the paper justified?

Yes

Should the paper be seen by a specialist statistical reviewer?

No

Do you have any concerns about statistical analyses in this paper? If so, please specify them explicitly in your report.

No

It is a condition of publication that authors make their supporting data, code and materials available - either as supplementary material or hosted in an external repository. Please rate, if applicable, the supporting data on the following criteria.

Is it accessible?

Yes

Is it clear?

Yes

Is it adequate?

Yes

Do you have any ethical concerns with this paper?

No

Comments to the Author

Please see separate file (See Appendix A).

Review form: Reviewer 2

Recommendation

Accept as is

Scientific importance: Is the manuscript an original and important contribution to its field?

Good

General interest: Is the paper of sufficient general interest?

Good

Quality of the paper: Is the overall quality of the paper suitable?

Good

Is the length of the paper justified?

Yes

Should the paper be seen by a specialist statistical reviewer?

No

Do you have any concerns about statistical analyses in this paper? If so, please specify them explicitly in your report.

No

It is a condition of publication that authors make their supporting data, code and materials available - either as supplementary material or hosted in an external repository. Please rate, if applicable, the supporting data on the following criteria.

Is it accessible?

Yes

Is it clear?

Yes

Is it adequate?

Yes

Do you have any ethical concerns with this paper?

No

Comments to the Author

In this paper, the authors focus on the problem of automatically extracting (genera, age) relationships from scientific articles / books. Specifically, the authors developed a ML-based system using Snorkel.

The computer science aspect of this work is sound, and this work significantly extends previous work (e.g., Peters et al. 2014) by using more advanced ML techniques.

Decision letter (RSPB-2019-0022.R0)

19-Mar-2019

Dear Mr Kopperud:

Your manuscript has now been peer reviewed and the reviews have been assessed by an Associate Editor. The reviewers' comments (not including confidential comments to the Editor) and the comments from the Associate Editor are included at the end of this email for your reference. As you will see, the reviewers and the Editors have raised some concerns with your manuscript and we would like to invite you to revise your manuscript to address them.

When submitting your revision please upload a file under "Response to Referees" - in the "File

Upload" section. This should document, point by point, how you have responded to the reviewers' and Editors' comments, and the adjustments you have made to the manuscript. We require a copy of the manuscript with revisions made since the previous version marked as 'tracked changes' to be included in the 'response to referees' document.

Research ethics:

Use of animals and field studies:

Insufficient sharing of data can delay or even cause rejection of a paper.

All supplementary materials accompanying an accepted article will be treated as in their final form. They will be published alongside the paper on the journal website and posted on the online figshare repository. Files on figshare will be made available approximately one week before the

accompanying article so that the supplementary material can be attributed a unique DOI. Please try to submit all supplementary material as a single file.

Please submit a copy of your revised paper within three weeks. If we do not hear from you within this time your manuscript will be rejected. If you are unable to meet this deadline please let us know as soon as possible, as we may be able to grant a short extension.

Best wishes,
Professor John R. Hutchinson, Editor
Proceedings B
mailto:proceedingsb@royalsociety.org

Associate Editor
Comments to Author:

Many thanks for your interesting article. Both reviewers were very positive about the ms. Rev had 1 major issue which, when dealt with, will enhance the narrative surrounding the paper - an explicit contrast in the temporal effort needed between your novel approach and just using the old way to extract the data. S/he also has a couple of other relatively minor points. I hope these reviews are helpful to you to improve your paper.

Reviewer(s)' Comments to Author:

Referee: 1

Comments to the Author(s)
Please see separate file.

Referee: 2

Comments to the Author(s)
In this paper, the authors focus on the problem of automatically extracting (genera, age) relationships from scientific articles / books. Specifically, the authors developed a ML-based system using Snorkel.

The computer science aspect of this work is sound, and this work significantly extends previous work (e.g., Peters et al. 2014) by using more advanced ML techniques.

Author's Response to Decision Letter for (RSPB-2019-0022.R0)

See Appendix B.

Decision letter (RSPB-2019-0022.R1)

02-Apr-2019

Dear Mr Kopperud

I am pleased to inform you that your manuscript entitled "Text-mined fossil biodiversity dynamics using machine-learning" has been accepted for publication in Proceedings B.

Open Access

Your article has been estimated as being 9 pages long. Our Production Office will be able to confirm the exact length at proof stage.

Paper charges

Sincerely,

Professor John R. Hutchinson, Editor

Proceedings B

Associate Editor:

Board Member

Comments to Author:

Congratulations to the authors for comprehensively addressing the relatively few and minor issues raised by the original reviewers. There is no need for further review - I recommend accepting the paper as is.

Appendix A

I assume I was asked to review this paper because I am a fossil taxonomic database author/tender. While I know something about compiling databases and looking for patterns in them, I know basically nothing at all about machine-learning. Hence, while I am happy to provide a review, I would emphasize that I lack any real basis to evaluate the central method in this paper. My comments should be weighted in light of that and more notice taken of any expert reviews in this regard.

In general, I am sympathetic to efforts to automate information extraction from taxonomic text. While one can bemoan a steady loss with scant replacement of taxonomic expertise, we are at the point where many groups lack even a critical mass of workers who could meaningfully evaluate the full literature, never mind contribute to it. The present paper makes a convincing case that meaningful signal can be detected using automated approaches.

What I've left grappling with after reading the paper, though, is to what extent this approach really saves time over simply manually compiling the information sought. The procedure seems to require a couple of people with taxonomic expertise, and they seem to be churning through rather a lot of tasks to teach/calibrate/whatever the machine-learning entity. The paper is mostly about how that's done, and secondarily about its efficacy (which seems quite impressive). I couldn't find really any discussion of how far ahead all of this might get us. There is justification in the introduction that, essentially, lone wolves making databases takes months or years or whatever. So, okay, fine. How much does the present method speed things up? It's not obviously to me that it really does, because it's really compiling fairly simple information and to my (certainly ill-informed) eyes it's taking quite a lot of effort to get it done.

It's true that really "doing" a full taxonomic compilation is a mammoth endeavour that, yes, can take literally years. But here, all that's needed is species/genera matched to time intervals. To carry out the procedure described in the paper, it's said (top of p. 9) that two humans "well-trained in understanding the literature in question" which I assume means "taxonomically competent" (if not then what does it mean?) had to spend time labelling ten and a half thousand "candidates." Judging from the scale of things represented in Fig. 3, couldn't they have spent this time, you know, *actually compiling taxa and ages from the primary literature*? That basic information is not hard or terribly time-consuming to compile. It's the broad suite of accompanying data and its various complications that make full taxonomic databasing murder. It would be interesting to hear some discussion of why having to have some level of human experts labelling ten and a half thousand entities is quicker than having them just, you know, compile the target data directly. I understand that the paper is largely a proof of concept. But if this level of effort is required to get a decent match of relatively simple associations between named age intervals and taxa (something Sepkoski was already able to do by himself for all marine genera), what level of manual calibration would be required to extend the machine-learning approach to the consideration of the more complex and varied sets of information associated with fossil taxa?

So all I'm saying is that there is a claim at the outset that you can't just go and get the information directly because it's too time consuming. But thereafter, exactly how time consuming getting machine-learning to compile names and ages versus simply having your humans, you know, compile names and ages doesn't seem to be addressed.

I was left very convinced that this method can capture taxic temporal patterns. I was left confused as to whether going this route was more efficient than just having humans harvest the relatively simple bits of information compiled. The relative amounts of effort would seem to be the crux of why to do something like this, and some discussion of this would be welcome.

Minor comments

In general, there doesn't seem to be much reported in the body of the paper about the actual

numbers involved. Numbers of papers considered are discussed on p. 6 (though they are not explicit). Thereafter there is mainly discussion of accuracy rates. It was difficult to get a sense for the exact taxonomic scope of the exercise.

line 63 - I don't believe you can really say that the target literature for exercises such as this is "burgeoning." Professional study of systematics and taxonomy is famously in crisis. Certainly my own work indicates that the rate of documentation of new fossil taxa in at least one major group has plummeted since the 1980s.

lines 261-264. I don't know that you can really make a blanket statement like this. The first paper listed in your References cited was published in *Science*, and was essentially about human error rates in data-compilations. It showed that Sepkoski had human error in 73% of his entries that were examined.

Appendix B

Dear Professor John R. Hutchinson, the Associate Editor, and Reviewers,

Thank you for handling and reviewing our manuscript. Our responses are in italics and blue below your comments.

Associate Editor

Comments to Author:

Many thanks for your interesting article. Both reviewers were very positive about the ms. Rev had 1 major issue which, when dealt with, will enhance the narrative surrounding the paper – an explicit contrast in the temporal effort needed between your novel approach and just using the old way to extract the data. S/he also has a couple of other relatively minor points. I hope these reviews are helpful to you to improve your paper.

We greatly appreciate your finding reviewers for a ms such as this one, which falls between different fields. We are glad that both reviewers were positive about our work. We have tried to improve the narrative surrounding our ms, which we hope is satisfactory, given RI's comments. We did so in a concise manner in the manuscript itself, but also wanted to answer RI's concern more closely in this letter.

Reviewer 1:

I assume I was asked to review this paper because I am a fossil taxonomic database author/tender. While I know something about compiling databases and looking for patterns in them, I know basically nothing at all about machine-learning. Hence, while I am happy to provide a review, I would emphasize that I lack any real basis to evaluate the central method in this paper. My comments should be weighted in light of that and more notice taken of any expert reviews in this regard.

In general, I am sympathetic to efforts to automate information extraction from taxonomic text. While one can bemoan a steady loss with scant replacement of taxonomic expertise, we are at the point where many groups lack even a critical mass of workers who could meaningfully evaluate the full literature, never mind contribute to it. The present paper makes a convincing case that meaningful signal can be detected using automated approaches.

What I've left grappling with after reading the paper, though, is to what extent this approach really saves time over simply manually compiling the information sought. The procedure seems to require a couple of people with taxonomic expertise, and they seem to be churning through rather a lot of tasks

to teach/calibrate/whatever the machine-learning entity. The paper is mostly about how that's done, and secondarily about its efficacy (which seems quite impressive). I couldn't find really any discussion of how further ahead all of this might get us. There is justification in the introduction that, essentially, lone wolves making databases takes months or years or whatever. So, okay, fine. How much does the present method speed things up? It's not to me that it really does, because it's really compiling fairly simple information and to my (certainly ill-informed) eyes it's taking quite a lot of effort to get it done.

We are delighted that R1 agrees that we have made a convincing case that we can retrieve a meaningful biodiversity signal using automated approaches. On re-reading our own writing, we realize that we had been unclear in certain statements, leading to some misunderstandings. See paragraphs below on "human annotators" and "time-saving". We have already mentioned that our pipeline can be used on other taxa with relative ease (see original Conclusion) but we have added a sentence at the end of the abstract to emphasize this sentiment.

It's true that really "doing" a full taxonomic compilation is a mammoth endeavour that, yes, can take literally years. But here, all that's needed is species/genera matched to time intervals. To carry out the procedure described in the paper, it's said (top of p. 9) that two humans "well-trained in understanding the literature in question" which I assume means "taxonomically competent" (if not then what does it mean?) had to spend time labelling ten and a half thousand "candidates." Judging from the scale of things represented in Fig. 3, couldn't they have spent this time, you know, *actually compiling taxa and ages from the primary literature?* That basic information is not hard or terribly time-consuming to compile. It's the broad suite of accompanying data and its various complications that make full taxonomic databasing murder.

Human annotators: In the original lines 192-193, we wrote that "Two human annotators, well-trained (and equally so) in understanding the literature in question, manually labelled 10,416 candidates, [...]" We suspect our language here was unclear, and that perhaps R1 assumed that the two human annotators are bryozoan taxonomic experts. In fact, they were not experts at all. We have modified this section to say "Two human annotators manually labelled 10,416 candidates, 1000 of which were labelled by both persons. These annotators were not taxonomists or bryozoologists, but had basic degrees in biology, understood the Linnean classification system, and had geological knowledge sufficient to recognize stratigraphic age units."

R1 also posed the question of why we did not instead spend time on "actually compiling taxa and ages from the primary literature?" We did, in a sense, do

so, in 'labelling' 10,416 candidates. However, this 'labelling' takes much less time than a manual approach. We have added a brief statement to "Gold labels" section of the paper: "This labelling of candidates in the context of retrieved sentences takes much less time than an approach that starts with reading through numerous papers or monographs to find the relevant data wherever it might occur in the text, and subsequently entering each separate taxon name and age." (see also paragraph below labelled 'Time-saving').

It would be interesting to hear some discussion of why having to have some level of human experts labelling ten and a half thousand entities is quicker than having them just, you know, compile the target data directly. I understand that the paper is largely a proof of concept. But if this level of effort is required to get a decent match of relatively simple associations between named age intervals and taxa (something Sepkoski was already able to do by himself for all marine genera), what level of manual calibration would be required to extend the machine-learning approach to the consideration of the more complex and varied sets of information associated with fossil taxa?

Time-saving: We realize on re-reading our ms that the parts that were automated versus the manual labelling were not well-demarcated in our writing. First, the 10,416 candidates were all retrieved in an automated fashion. This automated retrieval for training purposes is orders of magnitude faster than a human reading through hundreds of journal articles, chapters, and monographs. We added the word automated in three places in the section labelled "Sentence parsing & named entities" to clearly reflect this.

The 10,416 candidates that were manually labelled were isolated sentences that the annotators read. They did not read anything else. These 10,416 candidates were derived from 4058 unique sentences, which when concatenated are about as long as four 100-page monographs, even if the sentences themselves came from 467 monographs or shorter research papers. It isn't all that unusual for a fossil bryozoan publication to contain descriptions of 50 or more species occurring in several localities of slightly different ages.

The 10,416 labelled candidates were then used to train a machine-learning algorithm which can now probabilistically classify any candidate within the range of variation of those 10,416 it has "learnt" from, in a span of time that is again orders of magnitude shorter than what a human can do. This means that adding information from 800 more, or 8000 more, or 80000 more monographs to the occurrence data is essentially equally effortless for us now.

While it is true that Jack Sepkoski compiled ranges by hand, those data are not amenable to the types of analyses that one is able to do using Paleobiology Database style data. While we have presented (for simplicity) Sepkoski style range through data, statistical analyses based on the much richer occurrence

data (i.e. PBDB style data which takes much longer time to collect) can now be done for cheilostomes (but it is outside of the scope of this ms as we mention in original line 323). In addition, and perhaps even more importantly, the sources of data are completely trackable in our work, unlike Sepkoski's data (which we tried to track with limited success).

Specifically, Sepkoski's (2002) references for bryozoan genera are sometimes missing, "pers. comm." or are difficult to trace. We understand that Sepkoski (2002) was published posthumously with minor edits and thus not in the proper condition he presumably would have preferred. In contrast, however, our information extraction procedure demonstrates a much higher level of reproducibility: for any occurrence relation, we can refer to the very sentence from which it was extracted (original lines 146-148).

So all I'm saying is that there is a claim at the outset that you can't just go and get the information directly because it's too time consuming. But thereafter, exactly how time consuming getting machine learning to compile names and ages versus simply having your humans, you know, compile names and ages doesn't seem to be addressed.

I was left very convinced that this method can capture taxic temporal patterns. I was left confused as to whether going this route was more efficient than just having humans harvest the relatively simple bits of information compiled. The relative amounts of effort would seem to be the crux of why to do something like this, and some discussion of this would be welcome.

We are glad to hear that RI is convinced that our approach works for capturing taxic temporal patterns. We believe that RI's confusion is mainly due to our lack of clarity, which we hope we have improved with our edits scattered throughout the revised ms. Note that we have also reworded our conclusion section summarize our thoughts on some of the issues brought up by RI.

Minor comments

In general, there doesn't seem to be much reported in the body of the paper about the actual numbers involved. Numbers of papers considered are discussed on p. 6 (though they are not explicit). Thereafter there is mainly discussion of accuracy rates. It was difficult to get a sense for the exact taxonomic scope of the exercise.

As RI has noticed, this is a proof-of-concept paper and hence we have reported (for this ms) results only for cheilostomes. In the original lines 125-129, we did report the numbers underlined "This corpus consisted of over

10,000 bibliographic references, only some of which were likely to contain species occurrence data. We filtered those with accompanying PDFs, yielding more than 2000 documents in the English language; more than 800 of these contained fossil occurrences that were useful for our text-mining pipeline”

We now realize this may not be sufficient. Hence, we added the phrase in lines 107-109, “The cheilostomes are the most species-rich group of bryozoans for which there are currently about 4800 known extant members (Taylor and Waeschenbach 2015).”

In addition, we did say we labelled 10,416 candidates in the original ms, but we did not say how many unlabelled candidates were subsequently labelled by the automated pipeline to contribute data to Fig.3. Hence we added the sentences in lines 241-234, “We used the relation classifier to automatically label remaining candidates. As there are some duplicates among these remaining candidates, we estimate that there were about 10000 unique unlabelled ones.”

We hope that these new numbers suffice to inform the reader of the taxonomic scope.

line 63 - I don't believe you can really say that the target literature for exercises such as this is “burgeoning.” Professional study of systematics and taxonomy is famously in crisis. Certainly my own work indicates that the rate of documentation of new fossil taxa in at least one major group has plummeted since the 1980s.

We agree with the reviewer in that “burgeoning” was a poor choice of words. We have changed the paragraph to indicate that the body of fossil literature is still expanding, as opposed to “burgeoning”.

lines 261-264. I don't know that you can really make a blanket statement like this. The first paper listed in your References cited was published in Science, and was essentially about human error rates in data-compilations. It showed that Sepkoski had human error in 73% of his entries that were examined.

We agree with R1 that the statement was composed clumsily. What we intended to say is that many papers (e.g. Liow and Finarelli 2014 that we cited, original line 263) that are based on global compilations like the Paleobiology Database, do not model and usually do not even discuss human errors in database compilation. We are impressed by Adrain & Westrop's (2000) efforts in correcting stratigraphic and taxonomic errors. We have rephrased our sentence as “We also note that human error rates in data-compilations are rarely explored quantitatively and never modelled in

palaeobiological analyses on large datasets compiled from the literature (e.g. Liow and Finarelli 2014, Liow et al. 2015, but see Adrain and Westrop 2000 for an exceptional investigation of such errors).”

Reviewer 2:

In this paper, the authors focus on the problem of automatically extracting (genera, age) relationships from scientific articles / books. Specifically, the authors developed a ML-based system using Snorkel.

The computer science aspect of this work is sound, and this work significantly extends previous work (e.g., Peters et al. 2014) by using more advanced ML techniques.

We appreciate R2’s observation that our work ‘significantly extends previous work.’ We wish to clarify, however, that we did not use Snorkel (Ratner et al. 2017) for any machine-learning analysis. Rather, we used Snorkel primarily as a framework to create training data, as they have a convenient interface that lends itself nicely to candidates that consist of two spans of text tokens in a sentence, like ours.

That said, there are three main components in our procedure where we use machine learning. First, there is FastText (Bojanowski et al. 2017) for the word embeddings. Second, we used CoreNLP (Manning et al. 2014) for the Part-of-Speech and Dependency grammar. Third, we used Keras (Chollet et al. 2018) and Scikit-learn (Pedregosa et al. 2011) for the relation classification analysis. These technical details and their citations can be found in our SI.

References

Adrain, J. M., and Westrop, S. R. (2000). An empirical assessment of taxic paleobiology. *Science*, 289(5476), 110-112.

Bojanowski, P., E. Grave, A. Joulin, and T. Mikolov. 2017. Enriching word vectors with subword information. *Transactions of the Association for Computational Linguistics* 5:135–146.

Chollet, F., et al. 2015. Keras. <https://keras.io>.

Manning, C., M. Surdeanu, J. Bauer, J. Finkel, S. Bethard, and D. McClosky. 2014. The Stanford CoreNLP natural language processing toolkit. Pages 55-60 in *Proceedings of 52nd Annual Meeting of the Association for Computational Linguistics: System Demonstrations*.

Pedregosa, F., G. Varoquaux, A. Gramfort, V. Michel, B. Thirion, O. Grisel, M. Blondel, P. Prettenhofer, R. Weiss, V. Dubourg, J. Vanderplas, A. Passos, D. Cournapeau, M. Brucher, M. Perrot, and E. Duchesnay. 2011. Scikit-learn: Machine learning in Python. *Journal of Machine Learning Research* 12:2825–2830.

Ratner, A., Bach, S. H., Ehrenberg, H., Fries, J., Wu, S., and Ré, C. 2017. Snorkel: Rapid training data creation with weak supervision. *Proceedings of the Very Large Databases Endowment* 11:269-282.

Sepkoski, J. J. 2002. A compendium of fossil marine animal genera. *Bulletins of American paleontology* 363:1-560.

Taylor, P. D., & Waeschenbach, A. (2015). Phylogeny and diversification of bryozoans. *Palaeontology*, 58(4), 585-599.